# A cross-sectional analysis of podiatrist-initiated review processes after issuing prescribed foot orthoses

Luke Donnan *, Anna Horn, Emma Baker

School of Allied Health, Exercise and Sports Sciences, Faculty of Science and Health, Charles Sturt University, Albury, New South Wales, Australia

☯ These authors contributed equally to this work.

* ldonnan@csu.edu.au

## Abstract

### Background

Foot orthoses are widely used in clinical practice to treat foot, lower limb and back pathology. As published information guiding the clinical use of foot orthoses is scarce, the aim of this study is to profile the review processes used by practicing podiatrists after issuing an orthotic device.

### Methods

A cross-sectional observational study design formed the basis for a self-administered online questionnaire. The questionnaire was distributed through podiatry networks based in Australia.

### Results

Two-hundred and thirty-eight practicing podiatrists participated in this study. Ninety-seven percent of respondents indicated that they would recommend a review appointment after the initial fitting of an orthotic device. Forty percent ($n = 84$) of respondents scheduled the first review appointment four weeks after the initial fitting, while 33% ($n = 69$) preferred a two-week review period. A second review consultation was standard practice for 32% ($n = 68$) or respondents, and were typically scheduled either two (23%, $n = 12$) or four (38%, $n = 20$) weeks after the initial review consultation. Annual review of orthotic devices was recommended by 64% ($n = 123$) of participants in the study, while 19% ($n = 37$) would suggest that yearly reviews were scheduled only if required.

### Conclusions

Variation was identified in the orthotic review processes used by practicing podiatrists, although most respondents recommend a routine short-term review appointment for foot orthoses. It is not clear why practitioners adopt such varied approaches. In the absence of

**Data Availability Statement:** All relevant data are within the manuscript and its Supporting Information files.

**Funding:** We acknowledge financial support provided by Charles Sturt University. The funders

had no role in study design, data collection and analysis, decision to publish, or preparation of the manuscript.

**Competing interests:** The authors have declared that no competing interests exist.

any clear evidence on this topic, it may be that the differing approaches to patient review reflect different philosophical perspectives regarding patient management.

## Introduction

Foot orthoses (FO) are considered by some practitioners for the treatment and prevention of foot, lower limb and back disorders [1], typically with the intention of modifying loads on anatomical structures [2]. A wide range of options exist to manipulate foot function, ranging from prefabricated devices that can be purchased over the counter to custom devices that are manufactured based on a physical or digital impression of a person's foot [3]. Manipulation of foot position as a mechanical intervention is believed to influence symptoms by changing function of the foot, leg, hip, pelvis and thorax [1, 4–6], although the precise way in which these structures respond is still not fully understood [2].

Recent systematic reviews indicate the efficacy of FO when treating foot pathology is inconsistent [7–9]. While these publications do inform the clinical application of FO, each author contextualised their findings by acknowledging the variability that exists in FO construction and characteristics when comparing studies [7–9]. Much like FO prescription habits, the review processes that follow an initial FO fitting are shown to vary between practitioners [10]. This variation may reflect differing foci of the practitioner [10], potentially highlighting a practitioner interest in either symptom relief [4, 11–15] or biomechanical response [15, 16]. It is not clear if either of these measures are more important that the other, nor is it known whether these factors impact the FO review processes adopted by clinicians.

In the absence of standardised procedures guiding practitioners when issuing FO to patients, the aim of this study was to profile the review processes that have been adopted by practicing podiatrists after issuing FO. Specifically, the intention of this study is to determine whether consistencies exist in the way practitioners schedule orthotic review consultations, and what factors may influence the decision-making process of practitioners with differing levels of experience.

## Materials and methods

An online, self-administered questionnaire was used to obtain information regarding the orthotic fitting and review processes used by practicing podiatrists. Ethical approval was provided by the Charles Sturt University Human Research Ethics Committee (Protocol number: H19288). The initial page of the online questionnaire included a participant information sheet and a consent form (S1 File). If participants agreed to the terms outlined in the participant information sheet, consent was given by clicking 'Next' and proceeding to the survey questions.

### Questionnaire development

The questionnaire was developed by two qualified podiatrists (LD, EB) experienced in the prescription and review of FO in clinical practice. To establish face validity of the questionnaire a third qualified podiatrist (AH) evaluated the questions to confirm accuracy from a clinical perspective. Further scrutiny was provided by a colleague, not from a podiatry background, whose role within the institution was to provide questionnaire design support. They evaluated the questionnaire to identify and amend confusing, poorly articulated, or leading questions. Having adopted and 'open survey' approach, all survey data were collected using the

SurveyMonkey.com platform allowing all data to be collated automatically. As there were no incentives offered for participation multiple choice options were provided for most questions to reduce time required to complete the questionnaire. The selection of options for multiple choice questions was primarily informed by the clinical and theoretical knowledge of the authors. To ensure respondents were not limited by the response options provided, provisions for open ended responses were made in situations where the multiple choices provided were not exhaustive.

An invitation to pilot test a draft version of the questionnaire and provide feedback was accepted by 10 qualified podiatrists. Review of the pilot testing involved a combination of response analysis and verbal feedback from participants. The pilot process resulted in changes to the wording of three questions, changes to response options for seven questions, removal of six and addition of four questions. The authors agreed upon a twenty-five item questionnaire to address the research questions while limiting the time burden on participants (see S1 File for full questionnaire). To protect participant anonymity and encourage honest responses demographic information was limited to duration of practice and the University from which undergraduate qualifications were obtained. The 25 questions were arranged and presented in a specific order over 18 separate screens that would follow a logical workflow when issuing and fitting a pair of FO to a patient. A completion bar was located at the bottom of each page allowing participants to monitor their progress. Participants were able to return to previous questions while completing the questionnaire but were unable to review or modify their responses after submitting. As the purpose of this publication is to evaluate foot orthosis review procedures, results relating to 10 of the 25 questions from the questionnaire have been analysed. These include two demographic questions (questions one and two), two questions relating to FO prescription habits (questions three and four) and six questions investigating review and follow-up procedures (questions 14–17, 21 and 24). Questions four and 24 comprised three and five sub-questions, respectively, making for a total of 16 responses. The review-based questions placed an emphasis on the clinical decisions made pertaining to the use of review appointments, and the associated timeframes, following an initial FO fitting consultation.

## Recruitment

Using a convenience sampling approach the questionnaire was promoted through a variety of means to maximise the sample size and ensure diversity of podiatry experience and tertiary education backgrounds. The authors targeted Australian podiatrists by using existing podiatry contact lists, social media and official communications initiated by the Australian Podiatry Association, and private social media pages aimed at disseminating podiatry related information. All promotional communications included a direct link to the web-based questionnaire. An invitation to voluntarily participate in this study stipulated that prospective respondents were a qualified podiatrist. Therefore, qualification as a podiatrist justified inclusion in the study, while no podiatry qualification warranted exclusion from the study. Based on the promotional strategies used it was anticipated that most participants would be based in Australia, although prospective participants were not excluded based on geographical location.

## Data collection

The self-administered questionnaire was part of a cross-sectional observational study design. The questionnaire included predominantly multiple-choice questions using a nominal or ordinal scale. If the answers offered were not suitable participants could select 'other' to allow provision of an open-ended response. Contingency questions were used to ensure participants were

presented with questions relevant to their situation and responses. The questionnaire was available between September 27, 2019 and January 31, 2020. All data were stored within the Survey-Monkey platform during data collection before being downloaded as an excel file for analysis.

### Data analysis

Descriptive statistics were calculated in the form of frequencies and percentages. Proportional values were calculated relative to the number of actual responses for that question, not the total number of participants in the study.

## Results

### Demographic of respondents

A total of 238 podiatrists accepted the invitation to complete the questionnaire. Respondents included graduates from 10 current or former Australian podiatry courses, and institutions from New Zealand, South Africa and the United Kingdom. Most respondents graduated from Charles Sturt University (30%, $n = 71$), La Trobe University (18%, $n = 43$) and Queensland University of Technology (13%, $n = 31$) (Table 1). Ninety-two percent of respondents had been practicing for greater than one year, with the largest representation of practitioners having graduated between one and five years (31%, $n = 73$) or greater than 15 years (29%, $n = 69$) prior to completing the questionnaire. After 100% of participants attempted the first question (participation rate) a completion rate of 89% was observed (S1 Table). Calculation of response rates found 12 of 16 questions had response rates equal to or greater than 88% and all questions had a response rate greater than 75% (S1 Table). Furthermore, the use of IP addressed to identify unique participants indicated that all submissions were from different individuals. No questions were excluded from analysis.

**Table 1. Tertiary institution from which respondents obtained their podiatry qualification and years of practice.**

| Tertiary Institution | All | < 1 year | 1–5 years | 6–10 years | 11–15 years | > 15 years |
|---|---|---|---|---|---|---|
| | *n (%)* | *n (%)* | *n (%)* | *n (%)* | *n (%)* | *n (%)* |
| Auckland University of Technology | 7 (3) | 0 (0) | 3 (4) | 1 (2) | 3 (9) | 0 (0) |
| Charles Sturt University | 71 (30) | 10 (56) | 31 (42) | 15 (21) | 15 (21) | 0 (0) |
| Curtin University | 13 (5) | 0 (0) | 0 (0) | 0 (0) | 1 (1) | 12 (16) |
| La Trobe University | 43 (18) | 0 (0) | 9 (12) | 14 (19) | 6 (8) | 14 (19) |
| Queensland University of Technology | 31 (13) | 2 (11) | 9 (12) | 5 (7) | 6 (8) | 9 (12) |
| Southern Cross University | 1 (<1) | 0 (0) | 1 (1) | 0 (0) | 0 (0) | 0 (0) |
| Sydney Institute of Technology | 10 (4) | 0 (0) | 0 (0) | 0 (0) | 0 (0) | 10 (14) |
| University of Newcastle | 9 (4) | 1 (6) | 5 (7) | 3 (4) | 0 (0) | 0 (0) |
| University of South Australia | 18 (8) | 2 (11) | 7 (10) | 1 (1) | 1 (1) | 7 (10) |
| University of Western Australia | 6 (3) | 2 (11) | 3 (4) | 1 (1) | 0 (0) | 0 (0) |
| Western Sydney University | 10 (4) | 0 (0) | 3 (4) | 0 (0) | 1 (1) | 6 (8) |
| New Zealand Institution | 2 (1) | 0 (0) | 0 (0) | 0 (0) | 0 (0) | 2 (3) |
| South Africa Institution | 1 (<1) | 0 (0) | 0 (0) | 0 (0) | 0 (0) | 1 (1) |
| United Kingdom Institution | 8 (3) | 0 (0) | 0 (0) | 1 (1) | 1 (1) | 6 (8) |
| Not specified | 8 (3) | 1 (6) | 2 (3) | 2 (3) | 1 (1) | 2 (3) |
| **Total** | **238 (100)** | **18 (8)** | **73 (31)** | **43 (18)** | **35 (15)** | **69 (29)** |

*n* number of respondents in each category, % percentage of respondents from each tertiary institution proportional to the total number of respondents with equivalent years of practice experience. Percentage calculations do not include responses that are 'not specified'.

## Volume of prescriptions and types of orthotics

When asked how many pairs of orthotics were prescribed in a standard week, most respondents reported less than one pair (23%, n = 54), one to three pairs (39%, n = 91) or four to six pairs (21%, *n* = 49). Nine percent of respondents prescribed between seven and nine pairs in a standard week (*n* = 20), and fewer prescribed 10 or more pairs (7%, *n* = 16). Forty-nine percent (*n* = 107) of respondents indicated that they prescribed polypropylene devices more than 50% of the time, while 35% (*n* = 75) used predominantly ethylene-vinyl acetate devices for more than half of their orthotic prescriptions. Twelve practitioners (6%) prescribed carbon fibre orthotic devices more than 50% of the time (S2 Table).

## Scheduling of review consultations

Ninety-seven percent (*n* = 208) of respondents schedule a review consultation following the initial orthotic fitting, and 32% (*n* = 68) would schedule a second review consultation as standard practice (Table 2). Forty percent (*n* = 84) of respondents would schedule the first review consultation four weeks after the initial fitting, while 33% (*n* = 69) advise patients to return after two weeks (Fig 1 and S3 Table). Smaller proportions of respondents recommended a first review consultation after three weeks (12%, *n* = 24), six weeks (10%, *n* = 20) or eight weeks (2%, *n* = 4) (Fig 1). Of those practitioners scheduling a second review appointment as standard practice, the most common time frames were four weeks (29%, *n* = 20), two weeks (18%, *n* = 12), six weeks (13%, *n* = 9) and 12 weeks (9%, *n* = 6) after the first review consultation (Fig 2 and S4 Table). Sixty-eight percent (*n* = 142) reported that a second review consultation would only be scheduled if required (Table 2). Almost two-thirds (64%, *n* = 123) of respondents recommended a 12-month review of orthotic devices, with 15% recommending six monthly reviews (*n* = 28) and 19% advising reviews on a needs basis (*n* = 37) (Table 3).

## Factors influencing the scheduling of review consultations

When asked about factors that influenced orthotic review processes, most respondents suggested their professional judgement was the primary deciding factor (78%, *n* = 164). Most responses indicated patient preference (53%, *n* = 112) and laboratory factors (37%, *n* = 78) were moderately influential, and clinic protocols/employer preferences (45%, *n* = 96) and appointment availability (53%, *n* = 111) had minimal influence on the review schedule. Unlike any of the other experience-based groupings, those that had been practicing for less than one

**Table 2. Scheduling procedures adopted by respondents when performing foot orthoses review consultations.**

| Experience | All | <1 year | 1–5 years | 6–10 years | 11–15 years | >15 years |
|---|---|---|---|---|---|---|
| | *n (%)* | *n (%)* | *n (%)* | *n (%)* | *n (%)* | *n (%)* |
| *Do you schedule review consultations after the initial orthotic fitting?* | | | | | | |
| No | 7 (3) | 1 (7) | 2 (3) | 2 (5) | 0 (0) | 2 (3) |
| Yes | 208 (97) | 14 (93) | 66 (97) | 37 (95) | 32 (100) | 59 (97) |
| **Total** | **215 (100)** | **15** | **68** | **39** | **32** | **61** |
| *Would you schedule a second review consultation?* | | | | | | |
| Yes | 68 (32) | 2 (13) | 22 (33) | 11 (30) | 10 (31) | 23 (38) |
| Only if required | 142 (68) | 13 (87) | 44 (67) | 26 (70) | 22 (69) | 37 (62) |
| **Total** | **210 (100)** | **15** | **66** | **37** | **32** | **60** |

*n* number of respondents in each category, % percentage of respondents in each category proportional to the total number of respondents with equivalent years of practice experience.

**All participants**

| Weeks | 1 | 2 | 3 | 4 | 5 | 6 | 8 | 12 | 14 |
|---|---|---|---|---|---|---|---|---|---|
| n (%) | 3 (1) | 69 (33) | 24 (12) | 84 (40) | 2 (2) | 20 (10) | 4 (2) | 1 (<1) | 1 (<1) |

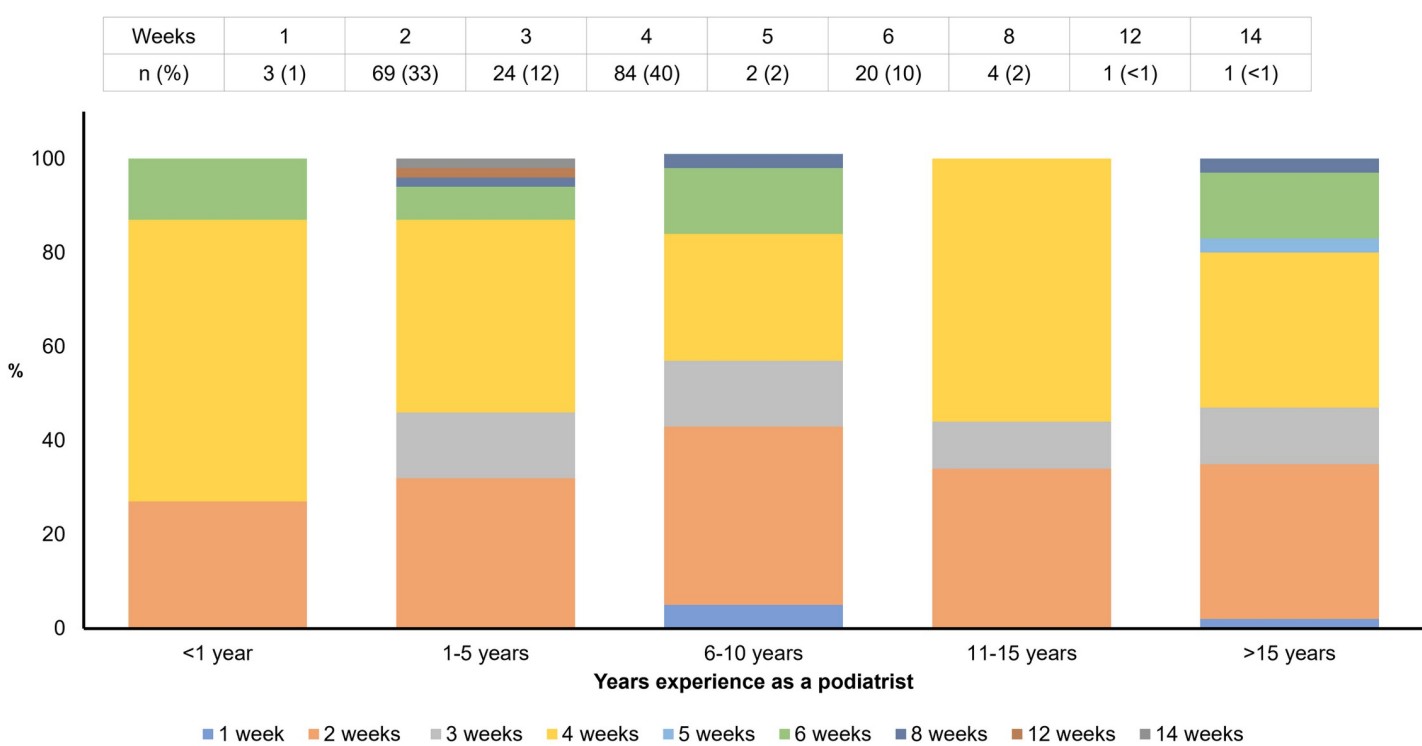

**Fig 1. How many weeks after the initial orthotic fitting would you schedule the first review consultation?.** Weeks: duration following an initial orthotic fitting consultation that a practitioner would schedule the first review consultation. *n*: number of respondents in each category. %: percentage of respondents proportional to the total number of respondents in each category.

year found patient preferences (60%, *n* = 9), clinic protocols or employer preferences (47%, *n* = 7) and laboratory-based factors (40%, *n* = 6) to be a highly influential factor when deciding on review processes (Fig 3 and S5 Table).

## Discussion

The purpose of this study was to profile the review processes used by practicing podiatrists after FO are dispensed. Almost all practitioners schedule at least one review appointment after issuing FO, yet the preferred duration between the issuing of the FO and the first review appointment did vary. These findings may reflect the absence of clear guidelines to direct a podiatrist's review processes, differing philosophical approaches between practitioners and the high reliance most practitioners place on their own professional judgement when determining review processes.

### Short-term review processes

Based on suggestions that a short-term review is six weeks or less [7, 15], 97% of practitioners in this study would conduct a short-term review after issuing a pair of orthotics. Forty percent of practitioners review four weeks after orthotic devices were issued, 33% of practitioners would schedule a two-week review, and three- and six-week reviews are the preference of 12% and 10% of practitioners, respectively. It is unclear what has precipitated these inconsistent appointment scheduling practices. A sample of studies assessing the efficacy of orthotic devices have used initial review periods of two [11, 17–19], four [13, 20] and six weeks [4, 14], typically

**All participants**

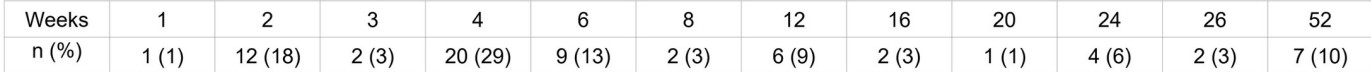

| Weeks | 1 | 2 | 3 | 4 | 6 | 8 | 12 | 16 | 20 | 24 | 26 | 52 |
|-------|---|---|---|---|---|---|----|----|----|----|----|----|
| n (%) | 1 (1) | 12 (18) | 2 (3) | 20 (29) | 9 (13) | 2 (3) | 6 (9) | 2 (3) | 1 (1) | 4 (6) | 2 (3) | 7 (10) |

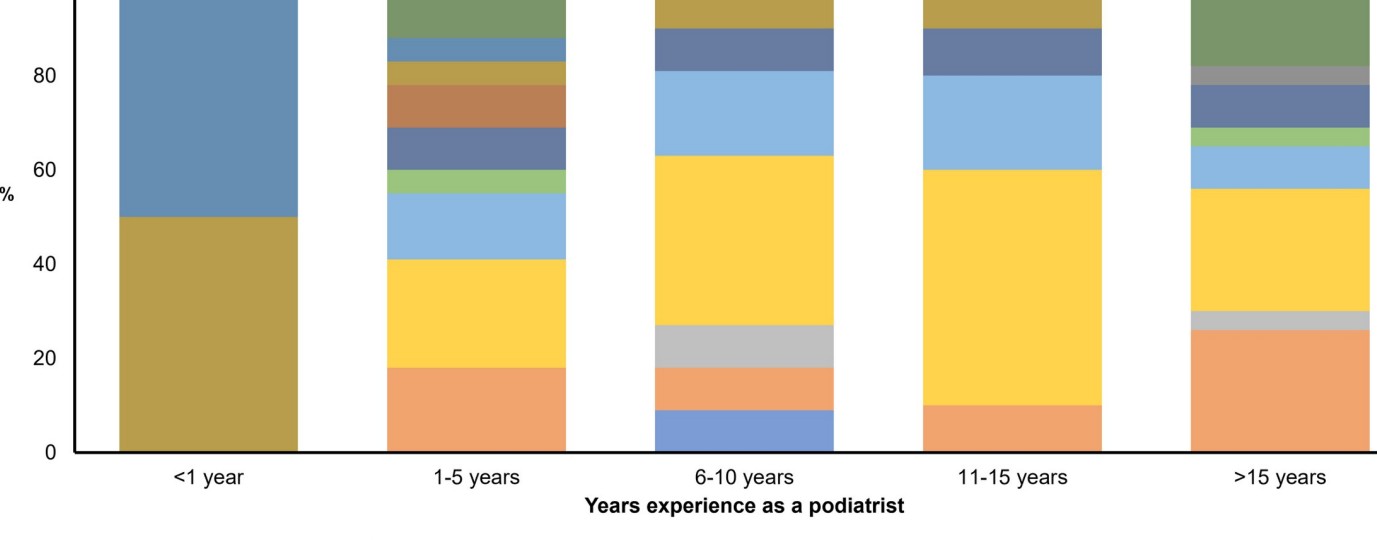

■ 1 week ■ 2 weeks ■ 3 weeks ■ 4 weeks ■ 6 weeks ■ 8 weeks ■ 12 weeks ■ 16 weeks ■ 20 weeks ■ 24 weeks ■ 26 weeks ■ 52 weeks

**Fig 2. How many weeks after the first review consultation would you schedule the second review consultation?.** Weeks: duration following the first orthotic review consultation that a practitioner would schedule the second review consultation. *n*: number of respondents in each category. %: percentage of respondents proportional to the total number of respondents in each category.

with no justification as to why those time frames were chosen. When assessing the impact of orthotic devices on functional ankle instability, Hamlyn and colleagues (2012) completed a two-week review based on the premise that each participant should increase the use of their orthotic devices by one hour every day, meaning the devices would be worn for a full day after two weeks [17]. Similarly, Gross et al. (2002) selected a two-week initial review period as this would allow participants one week to adjust to the devices, and one week for general use of the devices [19]. In both instances it would appear the preference for a two-week review period coincides with projections of when a patient should be able to wear their orthotic devices

**Table 3. Scheduling procedures adopted by respondents when performing a long-term foot orthosis review consultation.**

| | All | < 1 year | 1–5 years | 6–10 years | 11–15 years | > 15 years |
|---|---|---|---|---|---|---|
| | *n (%)* | *n (%)* | *n (%)* | *n (%)* | *n (%)* | *n (%)* |
| 6 months | 28 (15) | 2 (14) | 12 (19) | 3 (9) | 2 (7) | 9 (17) |
| 12 months | 123 (64) | 9 (64) | 41 (66) | 23 (66) | 18 (64) | 32 (59) |
| 24 months | 5 (3) | 0 (0) | 0 (0) | 1 (3) | 3 (11) | 1 (2) |
| On a needs basis | 37 (19) | 3 (21) | 9 (15) | 8 (23) | 5 (18) | 12 (22) |
| Other (please specify) | 21 | 1 | 6 | 4 | 3 | 7 |
| Total | 193 | 14 | 62 | 35 | 28 | 54 |

*n* number of respondents in each category, % percentage of respondents in each category proportional to the total number of respondents with equivalent years of practice experience. Percentage calculations do not include responses of 'Other (please specify)'.

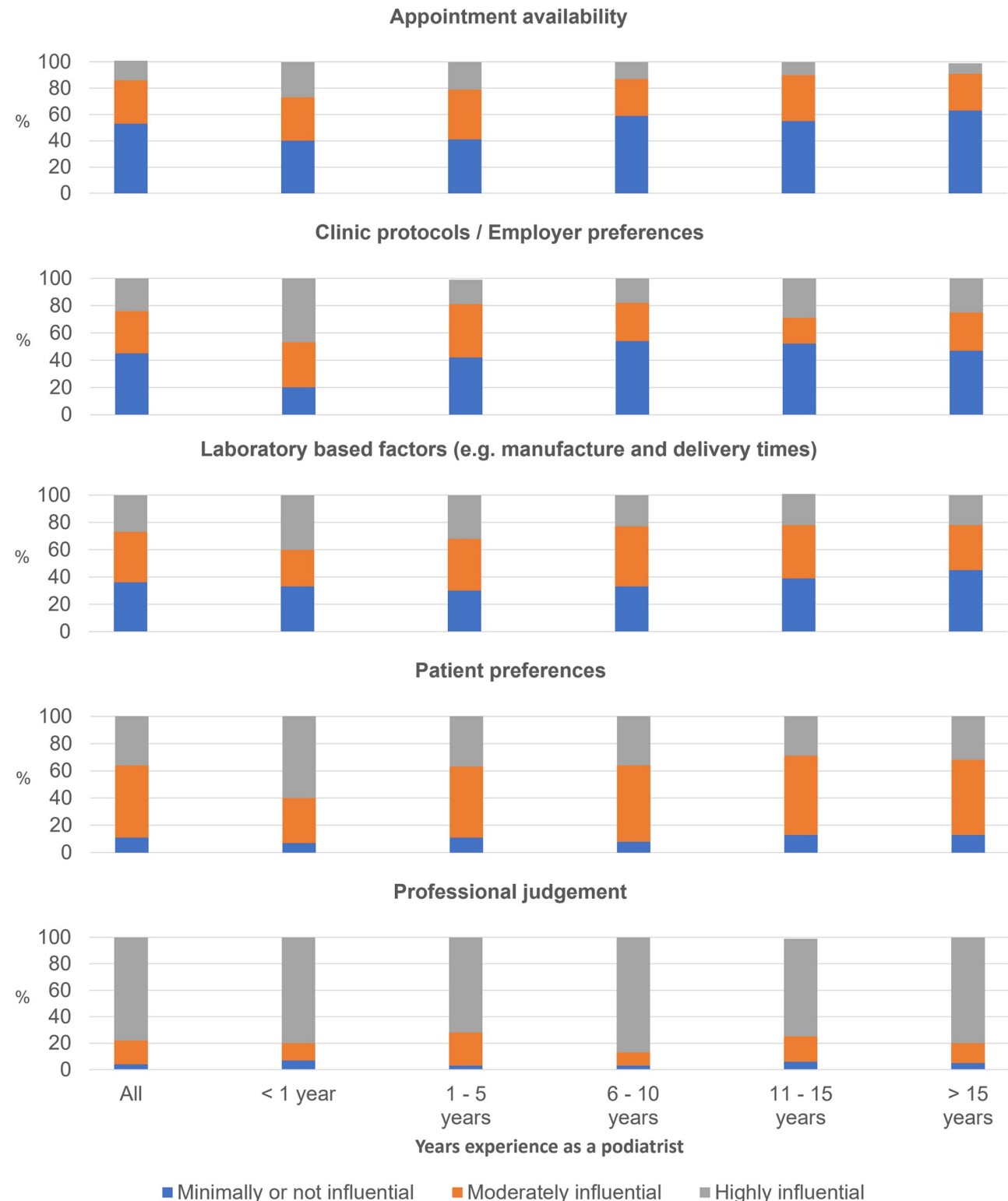

**Fig 3. How influential are the following on your orthotic issue and review processes?.** %: percentage of respondents proportional to the total number of respondents in each category.

comfortably for a full day. It may be that practitioners recommending a two-week review consider patient comfort a priority when looking to achieve symptom relief and reviewing immediately following completion of the wearing-in process may allow potential comfort-based issues to be addressed promptly.

The largest proportion of practitioners (40%) opted for a four-week initial review of prescribed orthotic devices. Collins et al. (2008) adopted a six-week initial review period when assessing the impact of orthotic devices on patellofemoral pain, stating that six weeks may have been the time of greatest treatment effect based or earlier studies focussed on patellofemoral pain [14]. Use of a six-week review has also identified reductions in lower back pain following orthotic intervention [4]. These examples of six-week reviews do not justify the choice of most podiatrists to review FO at four weeks. However, the use of a longer short-term review period may recognise symptom relief as a priority for these practitioners, and a four-week review period may allow adequate time for initial symptom changes to have occurred.

Assuming that two-week reviews are driven by comfort-based markers of progress, while four-week reviews could place a greater emphasis on symptom relief as a key indicator, the variation in short-term review processes may reflect differing practitioner philosophies. While these proposed justifications are merely speculative, if true, both philosophical positions focus primarily on patient satisfaction and not necessarily biomechanical correction. This interpretation would align with recent reporting that the success of an orthotic device was no longer determined based on theoretical biomechanical models, but instead, levels of patient approval [10].

## Long-term review processes

Sixty-four percent of podiatrists stated that they would schedule an annual review of prescribed FO, while 19% would review only on a needs basis. The practice of annual reviews is of interest as multiple orthotic-based studies have identified negligible differences between intervention and non-intervention at long-term review consultations [4, 14, 16]. When Cambron et al. (2017) compared the use of 1) no-intervention, 2) orthotic intervention or 3) orthotics plus chiropractic intervention, on lower back pain, all groups reported significant reductions in pain when reviewed at 12 months and no differences were observed between groups [4]. Similarly, management of patella-femoral joint pain using 1) flat insoles, 2) physiotherapy treatment, 3) prefabricated foot orthoses, and 4) a combination of physiotherapy and prefabricated foot orthoses, symptom improvement was noted among all groups at a 12 month follow-up, and no significant differences were noted between groups [14]. Comparable findings have also been reported when using sham, pre-fabricated and customised orthotics to manage plantar fasciitis [16], and when using combinations of heel raises, pre-fabricated orthotics and footwear to manage calcaneal apophysitis among paediatric patients [20]. While the studies cited consistently indicate that pain [14] and function [4, 16] improve more rapidly among the intervention groups, if the difference in symptoms between orthotic treatment and non-treatment groups appears unlikely to extend beyond a medium-term review, it may nullify the therapeutic value of annual orthotic review appointments.

## Variation between experienced and inexperienced practitioners

In the absence of clear evidence or guidelines relating to orthotic prescription and review procedures [21, 22] we can only speculate as to why practitioners have adopted such specific review processes. It may be that experienced practitioners typically use a pattern recognition model of clinical reasoning [23], whereby clinical judgements are based on previous experiences that resulted in successful outcomes [24]. Given that almost 70% of respondents identify their own professional judgement as being highly influential when scheduling review appointments, experienced practitioners may be informed primarily by their own anecdotal evidence,

which may be why individual results are so disparate. A lack of anecdotal evidence may also be why practitioners with less than one year of clinical experience reported variables such as patient preferences, clinic protocols, employer preferences and laboratory factors as being highly influential on their decisions regarding review processes. Without a bank of anecdotal evidence, novice practitioners tend to use a hypothetico-deductive reasoning approach: constructing a hypothesis, testing that hypothesis, and evaluating the response to the intervention [24]. Without experience to support their decision making, the management decisions of a novice practitioner can be influenced by a range of peripheral factors [24], which may have been affirmed by these findings.

The presence of limitations should be considered when interpreting the findings of this study. The use of an online web-link for data collection means it is possible that the questionnaire may have been forwarded to inappropriate or unintended subjects, and this population may be represented in the results. While possible, the risk of unintended respondents was minimised by directing most promotional material to known podiatrists (via e-mail) or podiatry specific groups (via social media). As demographic data were not collected in the questionnaire, it is assumed but not confirmed that the cohort recruited were predominantly Australian based podiatrists. Assuming those assessed in this study are predominantly podiatrists registered in Australia, recruiting 238 of approximately 5600 podiatrists (registered in Australia at the time of data collection) represents under five percent of the Australian podiatry profession. For that reason, the findings may not be representative of all Australian podiatrists. It is also noted that Australian podiatrists differ in their orthotic prescription habits when compared to podiatrists based in New Zealand and the United Kingdom [2, 22], meaning these results may be limited in their applicability to practitioners practicing outside of Australia. Furthermore, the potential influence of the practice setting, the condition being managed and the age of the patient were not controlled for in this study and may influence clinical decision making due to cost, cost responsibility and timeframes for review. While 18 different tertiary institutions were represented in this study, the sample was disproportionately swayed towards three institutions due to the convenience method of sampling used to recruit participants. Although it was not apparent when assessing the raw data, the academic institution of origin may have influenced the approach of some practitioners.

This study has identified variation in the orthotic review processes used among those podiatrists who responded, although it is not yet clear why practitioners adopt such varied approaches. The differing approaches may be associated with different philosophical perspectives regarding patient management, practitioner experience, or may have evolved from the limited research available in this space. Further investigation may be beneficial to understand why practitioners have adopted these approaches and if the review process has any impact on patient outcomes. The findings of this study may also provide insights into the clinical decision making of experienced practitioners prescribing orthotic devices, and the factors that influence novice practitioners still developing their clinical procedures.

## Supporting information

**S1 File. Questionnaire: Issuing and reviewing orthotic devices in podiatric practice.**
(PDF)

**S1 Table. Response rate for each question analysed in this study.**
(DOCX)

**S2 Table. Proportion of different types of foot orthoses prescribed relative to the practitioner's years of clinical experience.**
(DOCX)

**S3 Table. Scheduling procedures adopted by respondents when performing an initial foot orthosis review consultation.**
(DOCX)

**S4 Table. Scheduling procedures adopted by respondents when performing a second foot orthosis review consultation.**
(DOCX)

**S5 Table. Factors influencing foot orthoses issue and review processes and magnitude of their influence relative to the practitioner's years of clinical experience.**
(DOCX)

## Acknowledgments

The authors would like to acknowledge Gail Fuller from the Spatial Data Analysis Network (SPAN) at Charles Sturt University for her assistance in preparing and administering the online questionnaire.

## Author Contributions

**Conceptualization:** Luke Donnan, Anna Horn, Emma Baker.

**Data curation:** Luke Donnan, Anna Horn, Emma Baker.

**Formal analysis:** Luke Donnan, Anna Horn, Emma Baker.

**Funding acquisition:** Luke Donnan.

**Investigation:** Luke Donnan.

**Methodology:** Luke Donnan, Anna Horn, Emma Baker.

**Project administration:** Luke Donnan.

**Supervision:** Luke Donnan.

**Validation:** Luke Donnan.

**Writing – original draft:** Luke Donnan, Anna Horn, Emma Baker.

**Writing – review & editing:** Luke Donnan, Anna Horn, Emma Baker.

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
