## [Decision Letter · Decision Letter 0]

25 Aug 2022

PONE-D-22-22131A cross-sectional analysis of podiatrist-initiated review processes after issuing prescribed foot orthosesPLOS ONE

Dear Dr. Donnan,

Thank you for submitting your manuscript to PLOS ONE. After careful consideration, we feel that it has merit but does not fully meet PLOS ONE’s publication criteria as it currently stands. Therefore, we invite you to submit a revised version of the manuscript that addresses the points raised during the review process. 

Specifically1. Please respond to all comments raised by reviewer 1 and 2.2. Please align the reporting of your survey results with the Checklist for Reporting Results of Internet E-Surveys (CHERRIES).

We look forward to receiving your revised manuscript.

Kind regards,

Matthew Carroll, PhD., MEdL., MPod., BHSc

Academic Editor

PLOS ONE

Journal Requirements:

"The authors would like to acknowledge Gail Fuller from the Spatial Data Analysis Network (SPAN) at Charles Sturt University for her assistance in preparing and administering the online questionnaire."

"We acknowledge financial support provided by Charles Sturt University. The funders had no role in study design, data collection and analysis, decision to publish, or preparation of the manuscript."

Reviewers' comments:

Reviewer's Responses to Questions

**Comments to the Author**

1. Is the manuscript technically sound, and do the data support the conclusions?

Reviewer #1: Yes

Reviewer #2: Yes

2. Has the statistical analysis been performed appropriately and rigorously? 

Reviewer #1: N/A

Reviewer #2: Yes

3. Have the authors made all data underlying the findings in their manuscript fully available?

Reviewer #1: Yes

Reviewer #2: Yes

4. Is the manuscript presented in an intelligible fashion and written in standard English?

Reviewer #1: Yes

Reviewer #2: Yes

5. Review Comments to the Author

Reviewer #1: Thank you for the opportunity to review the manuscript titled ‘A cross-sectional analysis of podiatrist-initiated review processes after issuing prescribed foot orthoses’. While it fits the journal of submission, I would have thought a podiatry-focused journal may have been the target audience. However, whilst it is a relatively niche topic, the general principle related to duty of care, standardised practices and client centre management should have broad appeal. However again, given the journal audience, the rationale and presentation of outcomes of this survey could be stronger to enhance readership. The authors have covered the podiatry literature well but have stayed very podiatry-focused throughout. Perhaps identifying concepts such as ‘there is very little evidence on what FOs do, therefore the need to monitor their effect on clients is paramount’ – highlighting duty of care or alike may offer depth of premise as well as identifying that standards of care/guidelines need to be based on best practice principles and, in an absence of those, at least having an understanding of what is happening is a start. The argument for review times may also be strengthened by comparing to other professions – is there literature or guidelines for orthotists? Optometrists or any other profession that dispenses medical devices?

The writing style and the development of the manuscript has been well done. The authors should be commended on their efforts. I have made further suggestions below that may be helpful and identified a few things that require attention:

1. Please clarify ‘consent was implied’ – was consent gained specifically?

2. Where you have declared that ‘A third qualified author (AH) and an expert in online…’ Line 72, it would be helpful to put some references of previous work to show this author’s expertise.

3. Also good to identify how face validity was reviewed?

4. The reference for SurveyMonkey is needed (not the link to the platform itself)

5. Recruitment strategies via social media could be simplified to ‘social media associated with the Australian Podiatry Association or private pages aimed at disseminating podiatry related information’ rather than naming BTG.

6. I would recommend including a participant table in the manuscript (rather than as an appendices). It would be very beneficial to identify if the responding cohort represents the profession (e.g. compare to APHRA data).

7. As a suggestion, creating a table to display the main findings (including all participants regardless of experience) as your primary outcome alone with the impact of experience visually displayed via graphs might be beneficial. Having it all combined into the same tables means the reader needs to ‘seek’ results rather than have them displayed for them. A graph using a ‘stacked column’ approach would allow readers to see the impact of experience immediately and allow comparisons more readily.

Reviewer #2: Thank you for the opportunity to review this manuscript.

Abstract:

The statement on line 15 that "published information guiding the clinical use of foot orthoses is scarce" is broad and needs to be further clarified or supported.

I would suggest reviewing the way that numbers are presented throughout the manuscript as there is some inconsistency. For example on line 26, thirty-two percent should be represented as 32%.

Introduction:

Is the evidence to support the use of orthoses for back disorders strong enough to include it here? Similarly you mention that orthoses are believed to influence the function of the pelvis and thorax, is this supported by the evidence. The reference included on this statement by Cambron et al did not specifically investigate or discuss pelvic or thoracic biomechanics.

Methods:

You may benefit from clarification regarding the number of questions and specifically those questions that were omitted from this manuscript. I would suggest elaborating on what questions were excluded and why? Whilst I can understand your intention to focus this manuscript on orthotic review processes, that could have been a section of a more comprehensive review that may have provided further context for the findings. I can see in the survey within your appendix that questions which were not included relate to things like clinical assessment of orthotic suitability, patient education regarding wearing in the orthotic, adjunct interventions. These are all interesting questions which I believe would fit well into this manuscript. I do not see the benefit in removing these questions from this manuscript, I think the interest and clinical value of this study would be greatly improved if all questions were included.

Results:

You said that 238 registered podiatrists accepted the invitation. Do you have any way of confirming that they were registered? It may be better to just say podiatrists.

When reporting the volume of prescriptions I would suggest reordering these from less than one pair, one to three pairs, four to six pairs etc. I can see that it is currently listed in order of response rate but feel that it would be easier to follow if ordered by number of orthotics.

The total of people who prescribe polypropylene, EVA or carbon does not equal 100%. I understand that there are other materials considered such as PA11 and that these are included in your appendix. You may benefit from briefly listing these here.

Table 2. The totals do not add up for the question of how many weeks after you schedule your second follow up. Eg the total people who do use a second follow up is 68 people, yet there are only 52 responses to how many weeks later the session is. Is this data missing?

Discussion:

Line 207-209. Studies assessing the efficacy of orthotic devices have used a broad range of follow up times. Perhaps reword this sentence so that it is not portrayed as an exhaustive list.

Given the reliance on "professional judgement" that was found, it would be interesting to know what informed these judgements. For example, do these relate to pathology/diagnosis, biomechanics of the patient or objective, or other?

Line 247. This sentence would read better if begun with When Cambron et al compared the use of....

You had representation from 18 universities in the study, did this factor into the analysis, and if so, was a difference noted?

Line 303. You have stated in the limitations that these results may not be generalisable so should consider adjusting this sentence to read "this study has identified variation in the orthotic review processes amongst podiatrists who responded" or something similar.

I would be interested to know how you think these results will influence practice or future research? Maybe add a short section regarding future directions and research significance.

6. PLOS authors have the option to publish the peer review history of their article (what does this mean?). If published, this will include your full peer review and any attached files.

Reviewer #1: No

Reviewer #2: **Yes: **Aaron Jackson

---

## [Author Response · Author response to Decision Letter 0]

18 Sep 2022

The responses can be found in the attached document labelled 'Response to Reviewers'. The information has been itemised and presented in table form for ease of understanding.

---

## [Decision Letter · Decision Letter 1]

12 Oct 2022

A cross-sectional analysis of podiatrist-initiated review processes after issuing prescribed foot orthoses

PONE-D-22-22131R1

Dear Dr. Donnan,

We’re pleased to inform you that your manuscript has been judged scientifically suitable for publication and will be formally accepted for publication once it meets all outstanding technical requirements.

Kind regards,

Matthew Carroll, PhD., MEdL., MPod., BHSc

Academic Editor

PLOS ONE

Reviewers' comments:

Reviewer's Responses to Questions

**Comments to the Author**

1. If the authors have adequately addressed your comments raised in a previous round of review and you feel that this manuscript is now acceptable for publication, you may indicate that here to bypass the “Comments to the Author” section, enter your conflict of interest statement in the “Confidential to Editor” section, and submit your "Accept" recommendation.

Reviewer #1: All comments have been addressed

Reviewer #2: All comments have been addressed

2. Is the manuscript technically sound, and do the data support the conclusions?

Reviewer #1: Yes

Reviewer #2: Yes

3. Has the statistical analysis been performed appropriately and rigorously? 

Reviewer #1: Yes

Reviewer #2: Yes

4. Have the authors made all data underlying the findings in their manuscript fully available?

Reviewer #1: Yes

Reviewer #2: Yes

5. Is the manuscript presented in an intelligible fashion and written in standard English?

Reviewer #1: Yes

Reviewer #2: Yes

6. Review Comments to the Author

Reviewer #1: (No Response)

Reviewer #2: Thank you to the authors for making the detailed changes. I am satisfied that all comments have been addressed.

Thank you for clarifying the dissemination plan. I would suggest that this might be worthwhile briefly noting in the limitations section of this manuscript.

I look forward to reading the second half of this work once it is published.

7. PLOS authors have the option to publish the peer review history of their article (what does this mean?). If published, this will include your full peer review and any attached files.

Reviewer #1: **Yes: **Helen Banwell

Reviewer #2: No

---

## [Editor Report · Acceptance letter]

20 Oct 2022

PONE-D-22-22131R1 

A cross-sectional analysis of podiatrist-initiated review processes after issuing prescribed foot orthoses 

Dear Dr. Donnan:

I'm pleased to inform you that your manuscript has been deemed suitable for publication in PLOS ONE. Congratulations! Your manuscript is now with our production department. 

Kind regards, 

on behalf of

Associate Professor Matthew Carroll 

Academic Editor

PLOS ONE